# The Potentials and Pitfalls of a Human Cervical Organoid Model Including Langerhans Cells

**DOI:** 10.3390/v12121375

**Published:** 2020-12-01

**Authors:** Robert Jackson, Jordan D. Lukacs, Ingeborg Zehbe

**Affiliations:** 1Biotechnology Program, Lakehead University, Thunder Bay, ON P7B 5E1, Canada; 2Probe Development and Biomarker Exploration, Thunder Bay Regional Health Research Institute, Thunder Bay, ON P7B 6V4, Canada; izehbe@lakeheadu.ca; 3Department of Biology, Lakehead University, Thunder Bay, ON P7B 5E1, Canada; jdlukacs@lakeheadu.ca; 4Northern Ontario School of Medicine, Lakehead University, Thunder Bay, ON P7B 5E1, Canada

**Keywords:** cervical, keratinocytes, organoids, MUTZ cell line, Langerhans cells

## Abstract

Three-dimensional cell culturing to capture a life-like experimental environment has become a versatile tool for basic and clinical research. Mucosal and skin tissues can be grown as “organoids” in a petri dish and serve a wide variety of research questions. Here, we report our experience with human cervical organoids which could also include an immune component, e.g., Langerhans cells. We employ commercially available human cervical keratinocytes and fibroblasts as well as a myeloid cell line matured and purified into langerin-positive Langerhans cells. These are then seeded on a layer of keratinocytes with underlying dermal equivalent. Using about 10-fold more than the reported number in healthy cervical tissue (1–3%), we obtain differentiated cervical epithelium after 14 days with ~1% being Langerhans cells. We provide a detailed protocol for interested researchers to apply the described “aseptic” organoid model for all sorts of investigations—with or without Langerhans cells.

## 1. Introduction

Three-dimensional culture of human keratinocytes has seen an ongoing progress for more than four decades starting as in vitro skin grafting for clinical purposes [1]. Next-generation “organoids” not only contained skin components (keratinocytes, fibroblasts, and dermal collagen) but also immune cells to study the role of innate and adaptive immunity, e.g., lymphocytes [2] and epithelium-specific dendritic cells (DCs)—Langerhans cells (LCs) [3]. Regarding LCs, a technical advance with better reproducibility may be the bone marrow-derived cell line called MUTZ-3 [4] which can be matured into DCs and LCs using specific cytokine cocktails [1,5]. Currently, the Gibbs group seems to be making use of this approach most extensively, e.g., in the context of allergy and skin keratinocytes [5]. LCs typically make up approximately 1–3% of epithelial tissues [6], a ratio that is very precise [7].

Nevertheless, these methods used equally many or even twice the amount for organoids compared to the number of keratinocytes. The fact that not all cytokine-treated MUTZ-3 cells differentiate into CD1a^+^/CD207^+^ LCs has been the motivation for this undertaking. Indeed, a proportion of differentiated LCs could “switch back”, die or not attach, especially in an organoid scenario due to LCs maturing and migrating out of the epithelium before it is harvested. The obtained LC ratio in fully-grown organoids was not systematically investigated in any previous study. A comprehensive protocol is likewise lacking in the scientific literature.

Here, we describe a fully-detailed and reproducible procedure where the sub-group of LCs derived from MUTZ-3 cells is further “enriched” for seeding a more realistic proportion of myeloid cells (2 vs. 20% relative to keratinocytes) than reported above [7]. The outcome is a multi-layered, differentiated 3D organoid ready to study a wide spectrum of processes and functions in mucosal (e.g., cervical and oral) and skin tissues alike. While our focus is on human cervical epithelium, we describe an “aseptic” organoid tested for mucosa, as well as skin, with broad applicability, e.g., commensal microorganisms of healthy tissue; pathogens including bacteria, fungi, protozoan parasites, and RNA and DNA viruses, alone or in combination, as well as any other disease affecting mucosal and skin epithelia. Our research questions pertain to human papillomavirus (HPV) evolution and persistence leading to cancer, specifically regarding sub-lineages of type 16 [1,8,9]. This model, with or without varying ratios of LCs, can be used for investigations in health and disease where stratified epithelia are required for valid inferences.

## 2. Materials and Methods

### 2.1. Pre-Organoid Phase: Culture Concomitantly Two Primary Cell Types and Two Cell Lines

All cell types described herein are of human origin and were commercially obtained. Before growing organoid rafts with mucosal keratinocytes, cell culture has to be coordinated with the growth of primary cervical keratinocytes (Human Cervical Epithelial Cells (HCerEpiC); ScienCell Research Laboratories, Carlsbad, California, USA; Cat. #7060; no RRID number available), primary uterine fibroblasts (Human Uterine Fibroblasts (HUF); ScienCell Research Laboratories, Carlsbad, California, USA; Cat. #7040; no RRID number available), Langerhans cells (LCs) derived from *menschliche und tierische Zellkultur-3* (MUTZ-3) cells of an acute myeloid leukemia from a human male patient [Deutsche Sammlung von Mikroorganismen und Zellkulturen, Braunschweig, Germany; Cat. #ACC 295; RRID:CVCL_1433] and 5637 cells derived from a bladder carcinoma from a human male patient (American Type Culture Collection, Manassas, Virginia, USA; Cat. #HTB-9; RRID:CVCL_0126).

To generate the conditioned medium, 5637 cells (containing cytokines required for cultivating MUTZ-3) are grown in RPMI (Roswell Park Memorial Institute; medium traditionally used to grow lymphocytes)-1640 (Fisher Scientific, Cat. #A1049101), 10% fetal bovine serum (FBS) (HyClone, Cat. #SH3039603) and 1% antibiotics/antimycotics (AB/AM) (HyClone, Cat. #SV30079.01).

MUTZ-3 cells are grown in Dulbecco’s Modified Eagles Medium (DMEM) (HyClone, Cat. #SH3024301), 20% FBS, 20% 5637-conditioned medium, 1% AB/AM; for the progenitor population expansion and characterization: the number of MUTZ-3 cells needed for flow and the intended experiments, is based on subsequent cell counts and flow verification for determining the starting population. For the above experiments, we calculated a large starting population of 8 x 10^6^ MUTZ-3 cells, providing an excess in case they yielded only a low proportion of double-positive cells (e.g., 10%). The CD34/CD14 flow assessment of the MUTZ-3 cells is to verify their suitability for being progenitors of LCs (using PE mouse monoclonal anti-human CD34, clone 561, BioLegend, Cat. #343606 and FITC mouse monoclonal anti-human CD14, clone M5E2, BioLegend, Cat. #301804), as there are three subpopulation phenotypes within MUTZ-3 cells (each roughly a third of the total): the CD34^+^/CD14^−^ cells are the proliferative third, these can sequentially differentiate into CD34^−^/CD14^−^, which are a transitional third, and these can further differentiate into the non-proliferative “pre-LC” or “LC/DC precursor” third, which are CD34^−^/CD14^+^ [10]. Checking the initial population of MUTZ-3 progenitors is important as their phenotype changes during expansion; CD14^+^ and CD34^+^ populations should be approximately equal to start [10]. The differentiation process from MUTZ-3 to LCs (CD1a^+^/CD207^+^) takes 10 days and requires a cytokine cocktail (2.5 ng/mL TNFα, 100 ng/mL GM-CSF, and 10 ng/mL TGF-β1) in DMEM with 20% FBS and 1% AB/AM. LC ratio is then determined by flow cytometry (using PE mouse monoclonal anti-human CD1a, clone HI149, Novus Biologicals, Cat. #NB500-506PE and AlexaFluor 488 mouse monoclonal anti-human langerin/CD207, clone DCGM4/122D5, Novus Biologicals, Cat. #DDX0363A488-100). Since in our hands only~20% of the resulting cells are langerin^+^, further purification (enrichment) with magnetic microbeads (anti-CD207) (Miltenyi Biotec, Cat. #130-097-898) is done right before rafting, so the purified cells are viable and known to be CD207^+^. The process can be easily followed in the Miltenyi datasheet protocol for MS columns.

Concurrently with culturing MUTZ-3 cells: primary cervical keratinocytes are grown in EpiLife (Gibco, Cat. #LSMEPI500CA) with human keratinocyte growth supplement (HKGS) (Gibco, Cat. #LSS0015) and 1% AB/AM on collagen-coated T-75s (Corning Life Sciences, Cat. #C353136) as reported previously [11]; human dermal fibroblasts are grown in DMEM + 10% FBS + 1% AB/AM and expanded as described below or needed (Figure A1).

### 2.2. Organoid Rafting Procedure

Prior to beginning the organoid rafting procedure, the LCs must be grown first with conditioned medium from 5637 cells (as described above). Next, keratinocytes and fibroblasts are cultured. An example of growing organoids rafts with cervical keratinocytes is given. A suitable number of organoids, depending on the experimental design, should be prepared for morphological investigation, downstream molecular analyses, and bioinformatics. The steps are described in chronological order and illustrated in Figure A1.

This multi-step process requiring sterilized tools, preparation of several reagents and a good pair of steady lab hands, takes three days to prepare the organoids after which they have to grow for 14 days to mature to full-thickness epithelium.

Day 1: dermal equivalents are prepared in 48-well plates requiring 80,000 fibroblasts and 0.3 mL of rat tail-derived collagen (EMD Millipore, Cat. #08-115) per raft. Batch mixtures of dermal equivalents can be prepared based on the number of rafts using a small sterile glass beaker and stir-bar, surrounded by ice or cold beads on a stir-plate. Alternatively, for small preparations, a 50 mL conical tube on ice could be used and mixed carefully by hand. In either case, preparing additional dermal equivalents is recommended due to the high viscosity of collagen-containing solutions. To prepare the mixture in the cold glass beaker or tube, first add 0.3 mL/raft of collagen (diluted to 4 mg/mL in 0.02 N acetic acid). While mixing, 40 µL/raft of 10X Hanks’ Balanced Salt Solution containing phenol red (Millipore Sigma, Cat. #H4385) is added to indicate pH, followed by increments of 5 N NaOH (~1 µL/raft) to neutralize the solution (just enough to stay reddish-pink). Once neutralized, it is essential to work quickly while keeping the solution mixing (avoiding bubbles, however) and on ice, to prevent the collagen from prematurely solidifying. To the neutralized solution, add 80 µL/raft of fibroblasts in FBS (80,000 cells at 1 × 10^6^ cells/mL FBS) dropwise, while continuing to mix, and begin dispensing 400 µL/raft of mix to the 48-well plate. Incubate the plate for 20 min at 37 °C and 5% CO_2_ to allow gel solidification (opaque appearance). Carefully add 400 µL of fibroblast growth medium atop each dermal equivalent and return to the incubator overnight. The next morning, fibroblast elongation and colour change (from neutral to more acidic, red to orange) is observed. These are both good signs that the fibroblasts survived and are healthy.

Day 2: 250,000 keratinocytes/raft kept in 50 µL of medium are seeded on 48-well plates to form the basal keratinocyte layer, ensure they are homogeneously distributed across the top of the dermal equivalent by rocking the plate back and forth, allow 2 h attachment; seed LCs (e.g., 2 and 20% of an anticipated 500,000 keratinocytes upon an additional doubling after their initial seeding) kept in 50 µL of PBS (Fisher Scientific, Cat. #SH30028.02) and incubate for 30 min to improve attachment [12]; top up with 400 µL complete EpiLife medium to allow one more population doubling before differentiation medium is applied the next day (amounting to 500,000 cells before rafts are lifted). This calculation only applies to the further purified LC populations using magnetic beads with close to 100% double positive CD1a^+^/CD207^+^ LCs (i.e., experimental groups 2 and 3). Based on previous work [5], further purification like ours is not described which leaves us with only ~20% CD207^+^ LCs. Hence for experimental groups 4 and 5, we used 50,000 and 500,000 cells, respectively, of which the latter approximately equals their 1:1 epithelial-myeloid cell ratio.

Day 3: lifting to the air interphase is the most intricate part of the process requiring a steady hand and a small lifting spoon (e.g., Fisher Scientific, Cat. #21-401-15) to position the rafts, following careful aspiration of the medium on top, to round membrane inserts (Millipore Sigma, Cat. #PICMORG50) in 6-well plates; differentiation medium CnT-Prime 3D Barrier (CELLNTEC, Cat. #CNT-PR-3D) is now applied at 1.1 mL/well.

During the 14-day growth, medium changes occur every 2 days, and the collected medium is frozen to −80 °C (optional) to assess the supernatants for cytokines and growth factors if desired: 24 h prior to harvest, BrdU (10 µM final concentration) can be added to monitor proliferation (optional); also e.g., drug treatment or other manipulation can be administered as determined by study design.

### 2.3. Organoid Harvest

On day 15, remove medium followed by two PBS washes and then either fixing in 4% PBS-buffered formaldehyde (formalin) solution ON at RT followed by washing with and transferring into 70% ethanol to minimize protein over-crosslinking (morphological investigations with histological processing, hematoxylin and eosin staining, and various markers via in situ techniques, e.g., immunofluorescence (using a pan-cytokeratin mix of rabbit monoclonals anti-human cytokeratin 14, clone EPR17350, Abcam, Cat. #ab181595 and cytokeratin 19, clone EP1580Y, Abcam, Cat. #ab52625, and for LCs: mouse monoclonal anti-human langerin/CD207, clone 310F7.02, Novus Biologicals, Cat. #DDX0361P-100 and secondary) and immunohistochemistry as well as in situ DNA hybridization) or flash-freeze for molecular tests, e.g., DNA, RNA, and protein. We use the NucleoSpin^®^ TriPrep kit to simultaneously extract DNA, RNA, and protein (Macherey-Nagel, Cat. #740966); DNA is then stored at −20 °C; RNA and protein at −80 °C in 1.5 mL microcentrifuge tubes until used for subsequent assays. DNA/RNA purity and concentration are tested spectrophotometrically, e.g., with Nanodrop (PowerWave XS with Take3 Micro-Volume Plate, BioTek Instruments) and RNA integrity is tested with automated electrophoresis, e.g., Experion (Model #7007010, Bio-Rad Laboratories) before proceeding to downstream molecular analyses. Notably, RNA and especially DNA can also be extracted from formalin-fixed material, but some limitations such as fragmented nucleic acids have to be considered.

## 3. Results & Discussion

A framework for incorporating LCs into a keratinocyte organoid model is discussed in our recent publication [1], which revealed that incorporating these cells into the model is challenging. We wanted to avoid using an excess of LCs as this may hinder multi-layered epithelial differentiation and asked the following questions: (A) Does further enrichment of the CD207^+^ LCs allow us to use fewer myeloid cells than previously described? (B) How many myeloid cells are required to obtain a clinically relevant number of LCs in the harvested organoid? (C) What are the conditions to obtain properly differentiated organoids reminiscent of human cervical epithelium when interfacing with LCs?

We have optimized cytokine-induced differentiation of MUTZ-3 cells into LCs (CD1a^+^/CD207^+^, assessed via flow cytometry) and have attempted incorporating them into the organoid model. Experiments were performed using magnetic-enrichment to purify the differentiated MUTZ-3 (10–30% population of CD1a^+^/CD207^+^ when “unpurified”) and seeding different ratios of these cells relative to the primary human keratinocytes (2%, a realistic mean amount found in epithelia, assuming all embed vs 20%, 10× greater, in case there was loss). The same approach was performed using LCs without further purification. Altogether five experimental raft groups were grown. Group 1 contained cervical keratinocytes only; groups 2 and 3 contained purified (~98%) LCs at a ratio of 1:50 (10,000 LCs/500,000 keratinocytes) and 1:5 (100,000 LCs/500,000 keratinocytes), respectively; groups 4 and 5 contained unpurified (~20%) LCs at a ratio of 1:10 (50,000 LCs/500,000 keratinocytes) and 1:1 (500,000 LCs/500,000 keratinocytes), respectively.

The complex experimental design to grow organoids requires skilled lab hands, the ability to multi-task and coordinate lab work, organizing the substantial reagent stocks needed and the capacity to troubleshoot and find solutions. Knowledge of tissue morphology is essential as the detection of tissue-embedded LCs may be challenging for the inexperienced investigator. The time of LC seeding is crucial. We reckoned that it would be best to seed the LCs two hours after seeding the cervical keratinocytes as we know from previous experiments over the years of rafting that keratinocytes need approximately that time to adhere to the underlying dermal equivalents. Two independent proficient observers (R.J., I.Z.) calculated total keratinocyte yield using DAPI-stained nuclei as their criterion using the Northern Eclipse or ImageJ software counting feature. LC identification was pre-screened by R.J. and randomly confirmed by I.Z. A detailed overview of the pre-organoid phase and rafting procedure can be found in the Materials & Methods and Figure A1.

Optimal epithelial stratification was associated with increased purity and a lower ratio of LCs, as expected. Notably, tissue culture medium collections just before lifting the rafts to the air-liquid interface showed shed cells relative to the number of LCs seeded, but most for group 5 followed by groups 3, 4, and 2. Using 10-fold of the expected LC presence (group 3) yielded ~1% in the harvested rafts (i.e., within the normal range) rather than 20%. It could be possible that a large number of the LCs did not survive through the two-week rafting period (perhaps due to lack of optimal medium containing essential cytokines such as IL-4 and TNF-α). Using serial sections along with optimization of the immunostaining finally yielded their (rare) detection (Figure 1).

While we have developed the current epithelial organoid for modelling HPV infection to obtain insight into host innate immune defenses within the overall tissue microenvironment in dermal and epidermal compartments, it is also suitable for wider applications such as any type of pathogen-host relationships, general epithelial biology and personalized treatment testing. An extensive array of phenotypical and molecular tests on the single cell level or cell complexes for all cell types residing in the organoid can be performed, visualized, and analyzed with novel imaging tools in research and clinical settings. Results thereof can be matched and complemented with next generation sequencing (NGS) data and customized bioinformatics software triangulating qualitative and quantitative data. An important consideration for NGS is that the abundance of these keratinocyte immune markers may be very low and require sensitive methods i.e., deeper sequencing, as was the case in Jackson et al. [9], where the average read depth of ~40 million/sample was not enough to detect desired keratinocyte markers via RNA-Seq.

In Figure 2, we have suggested to elaborate on the study of signaling differences between keratinocytes and LCs for instance, specifically to elucidate the biology of keratinocyte and resident immune cell differentiation and function, but many other future directions can be explored in the context of the organoid’s intact morphology, molecular biology, imaging, and computer modeling. Prior to embarking on the complex organoid containing LCs, however, it may be worth considering that innate immunity can be assessed without LCs, using endogenous keratinocyte pathways and markers. Keratinocytes express a variety of relevant biomarkers that could provide insight into the innate immune environment and whether changes, e.g., due to viral activity could be immuno-evasive: innate sensing molecules such as TLRs, chemokines such as CCL27/28 and IL-18; pro-inflammatory cytokine expression such as TNF-α, TGF-β, and type I IFNs; cell-to-cell adhesion molecules, such as E-cadherin; and cell-surface receptors such as MHC I/II. Only a few works have considered the effect LC has on surrounding keratinocytes [13]. Interestingly, IL-15 signaling (Figure 2) may impact proliferation and induce anti-apoptotic effects—all aspects of wound healing, which an LC-free model could not capture. Future mucosal organoid research investigating IL-15 signaling in the context of microorganisms may yield new and unexpected results.

## 4. Conclusions

We sought to develop a reproducible immune-competent cervical organoid model with retained tissue integrity (e.g., differentiation). While we deem our described technical approach successful, due to the scarce presence of LCs in vivo (and in our model), it is debatable whether we will detect differences in the expression of the numerous immunomarkers expected in the HPV-infected cervical epithelium. To detect even subtle changes of such markers, a careful power analysis defining the appropriate sample size will be essential for our own and others’ investigations. Nevertheless, the fact that LCs are known to be relatively scarce in the cervical mucosa and that most HPV infections are cleared, it would be intriguing to be able to distinguish immune marker expression between their presence and absence in the engineered cervical organoid. Alternatively, LCs may not have the importance that we hypothesize, and may be minor actors in the HPV-infected epithelial microenvironment.

## Figures and Tables

**Figure 1 viruses-12-01375-f001:**
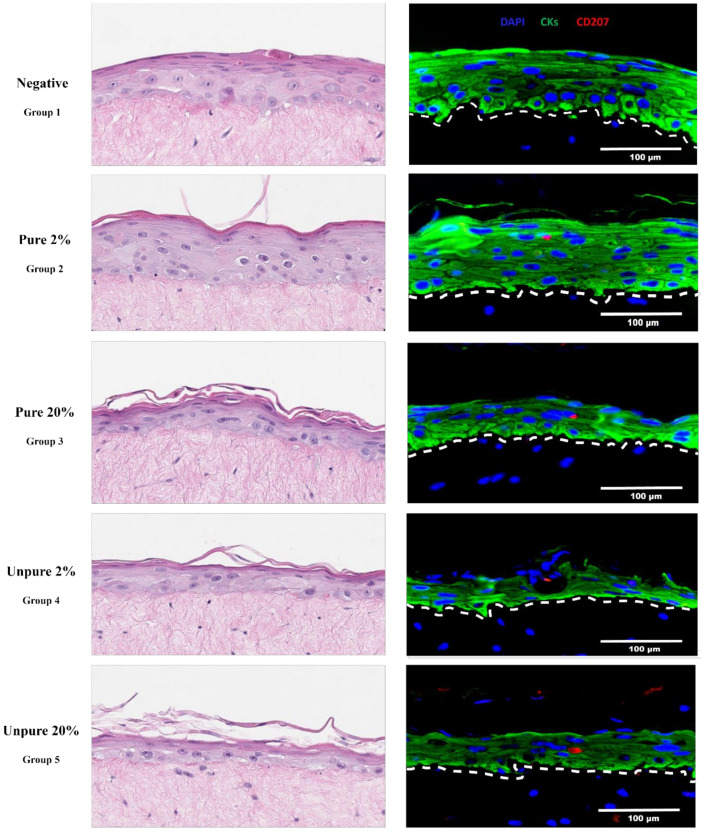
Immunocompetent cervical epithelium trial. Haematoxylin and eosin (H&E) stained micrographs (200× magnification, cropped) and immunofluorescence micrographs (400× magnification) of five cervical organoids with differential purity and ratios of Langerhans cells to keratinocytes. Nuclei are stained blue (via DAPI), green represents pan-cytokeratin immunostaining (CKs, a cocktail of cytokeratin 14 and 19, rabbit monoclonals anti-human cytokeratin 14, clone EPR17350, Abcam, Cat. #ab181595 and cytokeratin 19, clone EP1580Y, Abcam, Cat. #ab52625 with secondary antibody AlexaFluor 488 donkey anti-rabbit, Invitrogen, Cat. #A21206), and red represents langerin (CD207, mouse monoclonal anti-human langerin/CD207, clone 310F7.02, Novus Biologicals, Cat. #DDX0361P-100 and secondary antibody AlexaFluor 594 donkey anti-mouse, Invitrogen, Cat. #A21203). Using formalin-fixed and paraffin-embedded tissue, antigen retrieval had to be performed with Tris-HCl (pH 9) rather than citrate buffer (pH 6). Only then, could langerin^+^ (CD207) LCs be detected at a physiologically relevant level (~1%) in rafts of group 3. Groups 4 and 5 resulted in an LC detection rate of >1%, but at the cost of epithelial thickness. It should be noted that the langerin^+^ micrographs presented in this figure are not a complete reflection of the ratio that has identified in the organoids.

**Figure 2 viruses-12-01375-f002:**
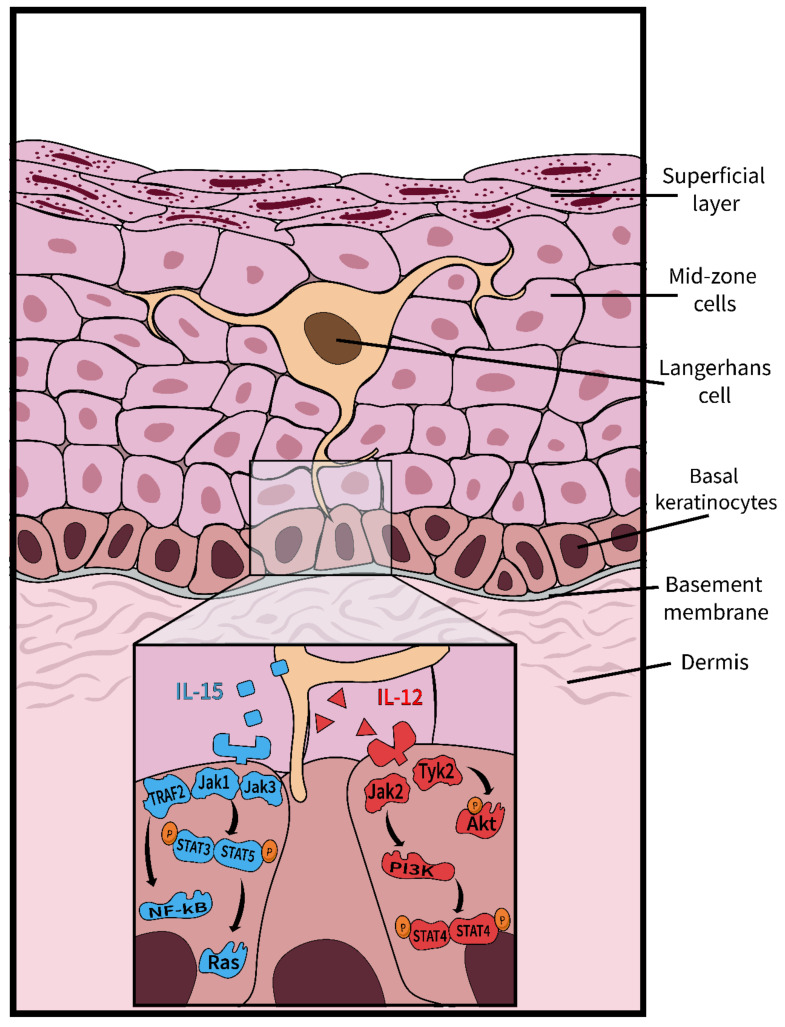
Langerhans cell signaling enhances the survival and proliferation of keratinocytes. Langerhans cells (LCs) that patrol the epithelium secrete two cytokines—IL-15 and IL-12—which are vital to the survival and proliferation of keratinocytes [14]. The IL-15 pathway (shown in blue) is initiated upon the binding of IL-15 to its respective receptor on the keratinocyte. The activated receptor induces the dimerization of STAT proteins, which in turn, activate the proteins Ras and Raf, known for their prominent roles in proliferation [13,15]. NF-kB can also be stimulated within this pathway, ultimately leading to the production of proinflammatory cytokines [15]. The IL-12 pathway (shown in red), activated upon the binding of IL-12 to the IL-12 receptor, triggers a phosphorylation cascade resulting in the phosphorylation of the Akt protein, a protein responsible for moderating cell survival [16]. A downstream effect of the IL-12 pathway is the production of IFN-γ, caused by the dimerization and phosphorylation of STAT proteins.

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
