# Peer review of "The Potentials and Pitfalls of a Human Cervical Organoid Model Including Langerhans Cells"

_viruses, 2020, doi:10.3390/v12121375_

Round 1

Reviewer 1 Report

The major problem with the manuscript is the use of the term organoid. At no point do the authors describe the establishment of what I would recognise as an organoid.  An organoid is a miniaturized and simplified version of an organ produced in vitro in three dimensions that shows realistic micro-anatomy. They are derived from one or a few cells from a tissue, embryonic stem cells or induced pluripotent stem cells, which can self-organise in three-dimensional culture owing to their self-renewal and differentiation capacities (Wikipedia). The technique the authors describe is one that we would recognise as a development of the organotypic raft approach, and the authors start with large numbers of keratinocytes. This technique was first described for use with HPV16 by Lambert and colleagues in 1999 (Establishment of the human papillomavirus type 16 (HPV-16) life cycle in an immortalized human foreskin keratinocyte cell line).

Author Response

Response to Reviewer 1 Comments

Point 1: The major problem with the manuscript is the use of the term organoid. At no point do the authors describe the establishment of what I would recognise as an organoid.  An organoid is a miniaturized and simplified version of an organ produced in vitro in three dimensions that shows realistic micro-anatomy. They are derived from one or a few cells from a tissue, embryonic stem cells or induced pluripotent stem cells, which can self-organise in three-dimensional culture owing to their self-renewal and differentiation capacities (Wikipedia). The technique the authors describe is one that we would recognise as a development of the organotypic raft approach, and the authors start with large numbers of keratinocytes. This technique was first described for use with HPV16 by Lambert and colleagues in 1999 (Establishment of the human papillomavirus type 16 (HPV-16) life cycle in an immortalized human foreskin keratinocyte cell line).

Response 1: Thank you for raising the point of terminology, as this is an important semantic issue that we have taken up in our recent work [Jackson et al., 2019], stating that there has been an evolution of 3D approaches (and terminology) including reconstituted skin, co-cultures, organotypic raft culture, and now “organoids”. We use the term “organoid” broadly to refer to 3D tissue culture models [Fatehullah et al., 2016], given its historical usage, which includes organotypic raft co-cultures. Throughout our Materials & Methods we specifically refer to the rafting approach as the method for generating these 3D cultures.

Jackson, R.; Eade, S.; Zehbe, I. An epithelial organoid model with Langerhans cells for assessing virus-host interactions. Philos. T. R. Soc. B. 2019, 374, 20180288. (doi:10.1098/rstb.2018.0288)

Fatehullah, A.; Tan, S.H.; Barker, N. Organoids as an in vitro model of human development and disease. Nat. Cell Biol. 2016, 18, 246-254. (doi:10.1038/ncb3312)

Reviewer 2 Report

The manuscript by Jackson and colleagues describes the preparation of a 3D cell culture model of human mucosal skin. They present a detailed protocol for the integration of Langerhans cells into the regenerating cervical epithelium as “organoids”. The described method using pure Langerhans cells is an improvement to the recently published protocols using unpure Langerhans cells. Organoids with pure Langerhans cells may represent valuable tools for basic and clinical research.

However, before publication I recommend to address the following issues, to improve the impact of the manuscript.

Major comments

  1. In order to show that the functionality of seeded Langerhans cells is not affected during epithelial regeneration at the air-liquid interphase, the authors should immunhistochemically stain the organoids (with unpure and pure Langerhans cells) for IL-12 and IL-15.
  2. Which effect do pure Langerhans cells have on keratinocyte growth in this model? Are differentiation and/or proliferation markers changed in this wound healing model to a pattern seen in human cervical tissue?
  3. The experimental procedure for flow cytometry and immunohistochemical staining could be described in more detail. Please include the antibody details.

Minor comments

  1. The passage from line 71 to 76 should be moved to section 2.2
  2. Why did they use conditioned 5637 media to cultivate MUTZ-3 cells?
  3. A schema for the pre-organoid phase described in 2.1 would help to understand the experimental procedure.
  4. The presented cartoon in Figure 2 needs to be labelled in more detail.

Author Response

Response to Reviewer 2 Comments

Note: we have highlighted changes in yellow as well as tracked changes in the revised manuscript's Word doc.

Point 1:
In order to show that the functionality of seeded Langerhans cells is not affected during epithelial regeneration at the air-liquid interphase, the authors should immunhistochemically stain the organoids (with unpure and pure Langerhans cells) for IL-12 and IL-15.

Response 1: Overall, we were primarily focused on describing a differentiation-based model, comprised of multi-layered epithelium, which is essential to study certain virus-host interactions (such as the differentiation-dependent viral life cycle of human papillomavirus). A wound healing model incorporating Langerhans cells (LCs) could be important for other groups, and in that context IL-12 and IL-15 secretion could be relevant and an area for future study, but we do not think staining would add to our current message.

Point 2: Which effect do pure Langerhans cells have on keratinocyte growth in this model? Are differentiation and/or proliferation markers changed in this wound healing model to a pattern seen in human cervical tissue?

Response 2: From the histological stains and the immunofluorescence (* Figure 1) we see multi-layered stratification of the cervical epithelium in the cultures containing pure LCs, including the basal cells (with proliferative capacity), suprabasal midzone/intermediate cells (differentiating upward), and the superficial differentiated cells of the upper-most layer. The stratification is optimal when pure and lower number of LCs are added (Pure 2%, Group 2), so we suspect the LCs are having a negative effect on the growth of the keratinocytes, but even in the Unpure 20%, Group 5, we still see fully-differentiated layers (albeit lower overall thickness), so they are not entirely inhibiting differentiation.

*Note: We noticed that the immunofluorescence panels within Figure 1 were pasted in the wrong order and have now ensured that these are correctly aligned and in the correct order corresponding to the histology stains in the updated Figure. The correct order is also in our pre-print version of this manuscript: https://www.biorxiv.org/content/10.1101/733501v4

Point 3: The experimental procedure for flow cytometry and immunohistochemical staining could be described in more detail. Please include the antibody details.

Response 3: We added the antibody details for flow cytometry (fluorophore-conjugated versions of CD34, CD14, CD1a, langerin/CD207) on lines 85 to 87 and 96-99. We added the formalin-fixed paraffin-embedded tissue-based immunofluorescence antibody details (pan-cytokeratins, CK14/CK19; langerin/CD207; and secondary fluorophore-conjugated antibodies) to lines 167 to 170, and in the Figure 1 caption, lines 228 to 233.

Point 4: The passage from line 71 to 76 should be moved to section 2.2

Response 4: We moved these lines to section 2.2 as recommended, with tracked changes.

Point 5: Why did they use conditioned 5637 media to cultivate MUTZ-3 cells?

Response 5: We use 5637-conditioned media which contains required cytokines to culture MUTZ-3 cells as per the recommendations of their supplier, DSMZ (Cat. #:ACC 295), and as described in their references [Drexler et al., 1997]. To provide more details to the readers: we added the catalogue numbers for 5637 and MUTZ-3 cells to the manuscript (Ln 69 and 71), as well as a note that the conditioned media contains required cytokines (Ln 73 to 74)

Drexler, H.G.; Zaborski, M.; Quentmeier, H. Cytokine response profiles of human myeloid factor-dependent leukemia cell lines. Leukemia. 1997, 11: 701-8. (doi:10.1038/sj.leu.2400633)

Point 6: A schema for the pre-organoid phase described in 2.1 would help to understand the experimental procedure.

Response 6: This is a great recommendation. We have revised Figure S1 to include the pre-organoid phase.

Point 7: The presented cartoon in Figure 2 needs to be labelled in more detail.

Response 7: We have revised Figure 2 to include additional label details, including the epithelial layers.

Round 2

Reviewer 2 Report

The authors have address my comments #2 to #7.

However, I still think it is important to show the functionality of seeded Langerhans cells as I stated in my comment#1, for example by immunhistochemically staining the organoids with unpure and pure Langerhans cells for IL-12 and IL-15. As the title of the manuscript describes the use of "Langerhans cells", it is absolutely necessary to proof that these Langerhans cells are functional. Organotypic raft cultures without Langerhans Cells have already been described before by other groups. In their recent paper (doi: 10.1098/rstb.2018.0288) the authors already describe the use of Langerhans Cells in organoids, therefore it should now be confirmed that these Langerhans cells are indeed functional.

Author Response

Point 1: However, I still think it is important to show the functionality of seeded Langerhans cells as I stated in my comment#1, for example by immunhistochemically staining the organoids with unpure and pure Langerhans cells for IL-12 and IL-15. As the title of the manuscript describes the use of "Langerhans cells", it is absolutely necessary to proof that these Langerhans cells are functional. Organotypic raft cultures without Langerhans Cells have already been described before by other groups. In their recent paper (doi: 10.1098/rstb.2018.0288) the authors already describe the use of Langerhans Cells in organoids, therefore it should now be confirmed that these Langerhans cells are indeed functional.

Response 1: Thank you for your comments on addressing the functionality of seeded Langerhans cells. The primary message we are trying to communicate with this article is how to facilitate the growth of 3D raft cultures by interested researchers, with a focus on the generation method and not directly on functionality or downstream applications. However, in addition to our immunostaining of Langerin/CD207 providing evidence that these embedded Langerhans cells are present, it also demonstrates that they are functional as they express the characteristic C-type lectin serving as antigen/pathogen uptake receptors in Langerhans cells (Valladeau et al., 2000; 2003). Overall, our manuscript and title is methodologically focused on the incorporation of Langerhans cells into 3D organotypic cervical epithelium, rather than on their downstream signaling, which could be taken up in future work. We would expect the interleukins are secreted in minute amounts which may not manifest well in an in situ staining of cut sections based on paraffin-embedded, formaldehyde-fixed tissue. Indeed, such investigations of Langerhans cell activity are generally done by flow cytometry for cell-based assays or ELISA using medium supernatant when growing cells or 3D rafts.

Valladeau J, Ravel O, Dezutter-Dambuyant C, Moore K, Kleijmeer M, Liu Y, Duvert-Frances V, Vincent C, Schmitt D, Davoust J, Caux C, Lebecque S, Saeland S. Langerin, a novel C-type lectin specific to Langerhans cells, is an endocytic receptor that induces the formation of Birbeck granules. Immunity. 2000 Jan;12(1):71-81. doi: 10.1016/s1074-7613(00)80160-0. PMID: 10661407.

Valladeau J, Dezutter-Dambuyant C, Saeland S. Langerin/CD207 sheds light on formation of birbeck granules and their possible function in Langerhans cells. Immunol Res. 2003;28(2):93-107. doi: 10.1385/IR:28:2:93. PMID: 14610287.

Round 3

Reviewer 2 Report

All my comments have been addressed.